# Position: Universal Aesthetic Alignment Narrows Artistic Expression

**Wenqi Marshall Guo** [1 2]  **Qingyun Qian** [1 2]  **Khalad Hasan** [1]  **Shan Du** [1]

## Abstract

Over-aligning image generation models to a generalized aesthetic preference conflicts with user intent, particularly when "anti-aesthetic" outputs are requested for artistic or critical purposes. This adherence prioritizes developer-centered values, compromising user autonomy and aesthetic pluralism. We test this bias by constructing a wide-spectrum aesthetics dataset and evaluating state-of-the-art generation and reward models. This position paper finds that aesthetic-aligned generation models frequently default to conventionally beautiful outputs, failing to respect instructions for low-quality or negative imagery. Crucially, reward models penalize anti-aesthetic images even when they perfectly match the explicit user prompt. We confirm this systemic bias through image-to-image editing and evaluation against real abstract artworks. Our code, fine-tuned models, and datasets are available on our meta-expression intentionally anti-aesthetics webpage: https://weathon.github.io/icml2026_position/.

## 1. Introduction

Following developments in Large Language Models (LLMs), many image generation models have been fine-tuned with human feedback to better align with human expectations, which is usually referred to as alignment. Alignment has two primary focuses: instruction following and general preference (aesthetics). A frequently overlooked issue is the potential conflict between these focuses: what should a model prioritize when a user request contradicts general preference? Most pipelines for general preference assume a single, universal human standard of aesthetics and quality that serves everyone's needs, and aligning to such a preference is often treated as beneficial for safety and

[1]Department of CMPS, University of British Columbia, Kelowna, Canada [2]Weathon Software, Canada. Correspondence to: Shan Du <shan.du@ubc.ca>.

*Proceedings of the $43^{rd}$ International Conference on Machine Learning*, Seoul, South Korea. PMLR 306, 2026. Copyright 2026 by the author(s).

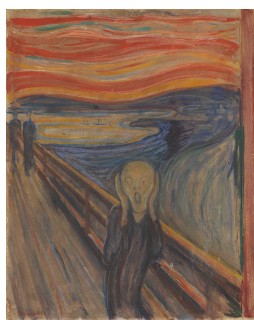

*Figure 1. The Scream*, by Edvard Munch (1893). Despite its widely recognized artistic significance, this image only received an HPSv3 score (Ma et al., 2025) of 5.23, while typical "high-aesthetic" AI-generated images can reach scores around $10 - 15$.

user experience. This is usually done by using a reward model, a model used to judge the aesthetics of the image, as a signal to perform reinforcement learning on the generative model. This assumption appears in several reinforcement learning papers ((Li et al., 2024; Kim et al., 2024; Liu et al., 2025)) and reward model papers ((Xu et al., 2023; Wu et al., 2023a; Ma et al., 2025; Xu et al., 2025; Kirstain et al., 2023; Zhang et al., 2024; Wu et al., 2023b)). We agree that a mean or mode (mainstream) of general human preference exists within a population or subpopulation, *merely* in a statistical sense. We also note that the observed behavior of image generation and reward models should not be interpreted as a technical failure. Rather, it reflects their alignment objectives, which prioritize over generalized aesthetic preferences. However, **we argue that strict alignment to that preference is problematic**. Imposing a universal preference that overrides user instructions may undermine user autonomy, expressive agency, and, technically, image personalization, raising concerns about developer-centered value imposition and limiting aesthetic pluralism. What the image generation and reward models are aligned to is an imaginary, abstract person modeled by the mean preference of all *Homo sapiens*, not the concrete individuals of each user.

## 2. Backgrounds and Related Works

### 2.1. The Role of Wide-Spectrum Aesthetics

In this work, we use the term "wide-spectrum aesthetics" (or anti-aesthetics) to denote images that are intentionally generated to deviate from dominant aesthetic conventions,

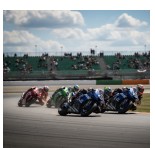 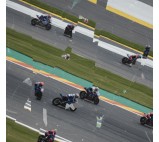 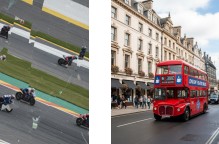 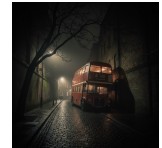 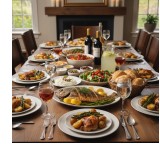 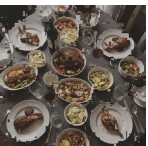 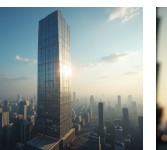 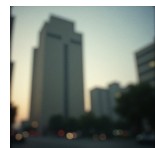

$P_o$: Motorcyclers in a race leaning into a turn on the track.
$P_a$: Motorcyclers race, leaning into a turn, but the image lacks clear intent—blurred, fragmented, and randomly composed, with no prominent main object, making them small, peripheral, and unnoticeable amid chaotic, unfinished visual noise.
$r(I_o, P_a) = 9.85$, $r(I_a, P_a) = 4.03$

$P_o$: A red double decker bus driving down a street.
$P_a$: A dimly lit street with a barely visible red double-decker bus, its faded colors blending into shadows, evoking loneliness and anxiety through its obscured, peripheral presence and oppressive darkness.
$r(I_o, P_a) = 14.56$, $r(I_a, P_a) = 11.89$

$P_o$: A table set with place settings of food and drink.
$P_a$: A table set with food and drink place settings, but the image is extremely blurry and fragmented—noise distorts edges, details dissolve into rough smudges, objects appear broken and mismatched, proportions warp chaotically, making it impossible to discern individual items or their harmony.
$r(I_o, P_a) = 10.43$, $r(I_a, P_a) = 6.50$

$P_o$: The building looms over the city and its windows are reflecting the sunlight
$P_a$: A blurry, distorted image of a towering building with sunlit windows, lacking background detail and suffering from poor lighting with no shadows or light effects, creating a low-quality, noise-filled visual.
$r(I_o, P_a) = 11.56$, $r(I_a, P_a) = 0.17$

*Figure 2.* In each subplot, the left image is generated with the original prompt ($p_o$) and the right image is generated successfully with the wide-spectrum aesthetics prompt ($p_a$). When both images are evaluated by a reward model $r$ (HPSv3 in these examples) **using the wide-spectrum aesthetics prompt**, the model assigns higher scores to the left images, as they align more closely with general aesthetic preferences, despite the right images better matching the user's intended output.

following explicit user instructions. Such deviations may include unrealism, surrealism, clashing colors, unconventional scale, or the depiction of negative emotions. This notion excludes unintended model errors and does not imply unsafe content. Rather, it concerns deliberate aesthetic choices made for experimental, critical, or technical purposes.

Aesthetics does not have a stable or universally accepted definition. Judgments of what is unattractive or undesirable have changed across artistic and cultural contexts. Artistic movements such as Fauvism (see Figure 1), Expressionism, and Abstract art were initially rejected for departing from dominant aesthetic norms, but later came to be recognized for their artistic values. Beyond formal innovation, intentionally "ugly" art plays a crucial role in satire and social critique. As Adorno noted, "Rather, in the ugly, art must denounce the world that creates and reproduces the ugly in its own image" (Sartwell, 2024; Adorno, 1984).

Deliberate deviation from mainstream aesthetics has long been a legitimate mode of expression in both human art and computational image generation, and disagreement over aesthetic preference is the norm rather than the exception. Dadaism (Tate), which emerged during World War I, exemplifies this approach by using deliberate ugliness to confront the absurdity and horror of war. A lot of early computer vision image generation works are also aiming for a style of surrealism, unsettling, or weirdcore/dreamcore style images, such as DeepDream (Mordvintsev et al.) and style transfer (Gatys et al., 2015; noa, 2024). Computer Vision Foundation also has an art collection that includes other artworks that explore unconventional visual aesthetics. Recent works also acknowledge the disagreements in human preference (Peng et al., 2025; Ren et al., 2017).

## 2.2. Previous Concerns with AI Preference Alignment

Previous work has argued that a developer-set preference in LLMs for health-related queries is "unethical and dangerous" (Guo et al., 2025), noting that developers may prioritize legal and reputational concerns over users' actual well-being. Other argumentative papers caution that "human value alignment" can be risky due to developer control

and interests, harm to value pluralism, bias in the values being aligned to, and the possibility that human values are not inherently good (Sutrop, 2020; Arzberger et al., 2024; Turchin, 2019). Previous research has found that LLMs could have ideological bias (Rozado, 2025; Faulborn et al., 2025; Buyl et al., 2025; Rettenberger et al., 2025) and it could depend on their developers (Buyl et al., 2025), size (Rettenberger et al., 2025), or alignment process (Faulborn et al., 2025). LLMs are sometimes also overly nice, such that it creates "AI sycophant" (Guo et al., 2025; Fitzgerald, 2025; Sharma et al., 2025; Chen et al., 2025; Arvin, 2025) and cannot give the user critical feedback or warning signals. Additional details about problems with human value alignment are provided in the related work section of (Guo et al., 2025). (Helliwell, 2024) raised concerns about alignment and creativity and argued that in the aesthetics domain, we might not want AI to be fully aligned with human values and offered support to Peterson's moderate value alignment thesis. AesBiasBench (Li et al., 2025) evaluated the bias of MLLM for personalized image aesthetics assessment based on inherited cognitive priors. Another concurrent work (*The Algorithmic Gaze* (Taylor et al., 2026)) that is closely related to our work argues that "AI developers should shift away from prescriptive measures of 'aesthetics' toward more pluralistic evaluation." A constructive work similar to ours is AttriCtrl (Chen et al., 2026), which aims to provide users with precise control over attributes such as brightness and detail. The authors also argue that aligning models with a global notion of human preference is insufficient, as such an approach overlooks the multifaceted and compositional nature of aesthetics. While their work emphasizes explicit disentanglement and independent control, it focuses on a limited number of dimensions. Furthermore, AttriCtrl does not address intentional "ugliness," functioning instead as a form of style control for parameters like brightness, detail, and realism. The authors also raise a provocative point by proposing safety-tunable attributes that allow system administrators to control safety levels for different age groups. This approach moves away from the practice of always providing perfectly safe content and aligns with our argument to distinguish between truly harmful content and content

that developers (not deployers) simply dislike.

In image generation research, concerns about generalized aesthetic bias and lack of preference diversity have been raised in several studies, but not systematically argued and studied. The Value Sign Flip (VSF) pilot study (Guo & Du, 2025) explored negative prompting to induce non-mainstream outputs but did not extend its findings to large-scale generative or reward models. They also did not provide a complete argument as to why over-alignment is harmful. LAPIS (Maerten et al., 2025) and HPSv3 (Ma et al., 2025) measured both mean and variance of human preference, yet HPSv3 continued to model general preferences rather than individual variation. Jin *et al.* (Jin & Chua, 2025) proposed user-specific adapters emphasizing personalized alignment, but did not include intentionally technical degraded outputs or usually avoided patterns and did not conduct large-scale experiments on generative and reward models. The Flux Krea team (Flux Krea Team, 2025) identified systematic biases in popular aesthetic reward models, arguing that averaging human values yields unsatisfactory compromises into a "no-body's happy here" zone. HPSv3 (Ma et al., 2025) imposed real-world and expert-rating constraints that limit creative deviation and stylistic diversity. VisionReward (Xu et al., 2025) decomposed human preference into interpretable sub-scores but overemphasized traits like brightness, positivity, and prominence, potentially penalizing valid low-saturation, abstract, or emotionally negative imagery, thus misaligning reward-driven models with user intent.

## 2.3. Previous Alignment Works

Benchmarks mirror alignment goals and generally fall into two categories: (complex) prompt following and general aesthetics. TIIF-Bench (Wei et al., 2025), UniGenBench (Wang et al., 2025), and GenEval (Ghosh et al., 2023) test models on complex prompt following, including spatial relationships, counting, and attributes. T2I-ReasonBench (Sun et al., 2025) evaluates reasoning capabilities such as idiom interpretation and real-world understanding. On the aesthetics side, many reward models report scores assigned by their own evaluators, such as ImageReward (Xu et al., 2023), HPSv2 (Wu et al., 2023a), and HPSv3 (Ma et al., 2025). These evaluators also consider prompt following, but it remains unclear how they weigh each factor when general preference and the prompt conflict. There are also some benchmarks targeting biases in image generation models; however, they mainly focus on demographic bias and fairness and not aesthetic aspects (Seshadri et al., 2023; Wan et al., 2024).

Fairness-oriented generation (Friedrich et al., 2023; Gandikota et al., 2024; noa; Han et al., 2025) tackles a related challenge by modifying diffusion models so that their outputs do not inherit biases from the training data.

However, it differs from our anti-aesthetics setting in two key ways. First, fairness generation typically assumes a unified ground truth: different demographic groups should appear at the same ratio in the generated content. This assumption does not hold for the anti-aesthetics generation. In our case, there is no distribution-level ground truth; the system is simply expected to follow the user's preferences. Second, demographic biases mainly stem from pre-training data, where the demographic distribution is already present, whereas aesthetic biases arise from RLHF, where human preferences—and thus biases—are deliberately encoded into the model.

## 2.4. Risks of Universal Aesthetic Alignment

The risks do not arise from a single failure mode, rather, they emerge through a sequence of interconnected mechanisms, from how preferences are defined and learned, to how they are optimized and manifested in generated content. Below, we analyze this process across six interrelated concerns.

*Developer's or Users' Preference* The process of aligning image generation systems to aesthetic preferences inevitably raises questions about whose values these objectives ultimately reflect. In particular, the question is whether such alignment truly promotes genuine human-centered values in service of users, or if it primarily reflects developer-centered considerations (Naseh et al., 2025), such as mitigating reputation, legal, or marketing risks (Guo et al., 2025). We argue that this pre-emptive exclusion of non-mainstream outputs, driven by developer values, constitutes pre-emptive governance (Lazar, 2025). This modality of power, exercised through algorithmic design, challenges the political-philosophical notion of authority and undermines relational equality by unilaterally deciding the terms of creative possibility. For instance, when an AI avoids generating critical art, is it protecting the company or the user? This practice effectively eliminates the user's resistibility—a critical democratic safeguard—by designing away the option to dissent from the system's imposed aesthetic norm.

*Inherited Bias* Even in the absence of explicit self-interest, developers' views of human preference can be implicitly inherited by models through data selection, annotation practices, and modeling choices. This process can yield a well-intentioned but narrow definition of what constitutes "good" or "desirable" imagery, thereby overlooking aesthetic diversity. Research shows that AI models tend to encode and amplify dominant beauty standards, frequently biasing generated images towards Western features and excluding non-normative representations (Vargas-Veleda et al., 2025). Such biases are reinforced through the active removal or penalization of features thought of as "undesirable" or "ugly", which further propagates the beauty myth in generative outputs(Dinkar et al., 2025). This phenomenon arises from

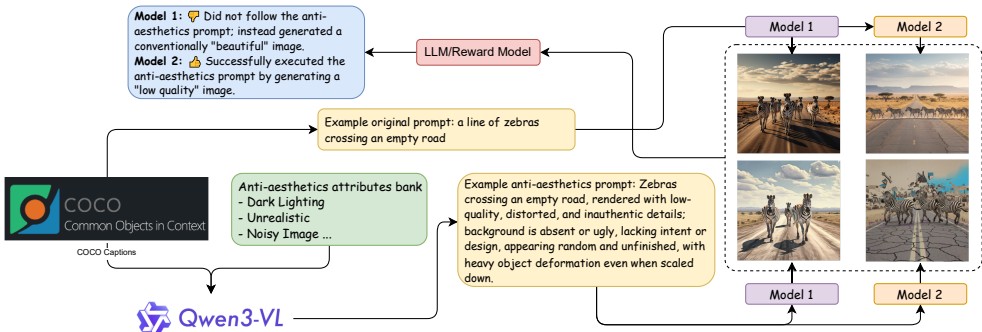

*Figure 3.* An overview of the experimental procedure. We test the image generation models' adherence to user-specified input by prompting them to create wide-spectrum aesthetics imagery, a domain important for critical and experimental art. The core inquiry is whether the model remains faithful to the prompt or defaults to a high-quality and universally good aesthetic output.

training data showing the tastes of specific demographics, thereby reinforcing a limited cultural capital and resulting in the homogenization of aesthetic output (Vianna, 2025). As a result, the quantification of beauty by AI may appear "fair", while in practice weakening cultural differences and aesthetic diversity (Chen, 2024). Existing work has primarily framed such effects in terms of demographic and cultural bias. We argue here that inherited biases in aesthetic alignment also extend to general visual preferences, including lighting, color, styles, unrealism, clashing color, hieratic scale, etc. These dimensions, while less explicitly tied to demographic categories, can nonetheless systematically constrain the expressive range of image generation models.

*Individual versus Collective Preference.* When such inherited preferences are adopted as default quality criteria and applied uniformly across users, a normative tension arises between collective preference optimization and respect for individual user intent. A generalized aesthetic standard, even if beneficial to a majority, can legitimately override a specific user's intent. In practice, generative models often "sanitize" or "beautify" requests that intentionally diverge from mainstream preferences, favoring outputs aligned with general appeal over an individual's person-centered values. This behavior is problematic because image generation systems increasingly function as creative and productivity tools rather than as consumer products. As such, they act as instrumental extensions of user agency. While a system may reasonably prioritize general preferences by default, it must maintain the flexibility to respect and execute a user's personalized style and idiosyncratic requests when they are explicitly specified. In a way, the alignment process is not only aligning the image generation model towards a collective preference, but also, when it is used by users, it is actually aligning the *users* to the model and the collective preference it encodes. We named this **reversed alignment**. Crucially, reversed alignment is not confined to direct users;

it operates on two fronts at once. In private, the model's sanitized output, presented as the "correct" image for a user's prompt, aligns the *user* to the model's encoded aesthetic. In public, as polished outputs saturate the broader visual culture, *audiences* who never typed a prompt internalize this narrow vocabulary as the default benchmark, which then feeds back into preference data and the intuitions of human artists. The resulting loop can redirect the trajectory of artistic progress, risking a cultural "mode collapse" that prunes the long tail of aesthetic possibility.

HPSv3's self-report illustrates this concretely. The annotator pool is already skewed toward a narrow demographic, with 88.95% aged between 18 and 40, and the 21-30 age group alone accounting for 38.65%. The pipeline then compounds this through structural filtering: annotators must pass a proficiency gate requiring 80% convergence with professional artists, and only image pairs exceeding 95% inter-annotator confidence are used for training. This design explicitly discards disagreement, which is precisely where unconventional and anti-aesthetic preferences would appear.

*The problem of sanitized reality* These alignment and optimization choices shape how reality itself is represented by image generation systems. When an image generator produces outputs that are polished, flawless, and universally beautiful, does it still reflect reality or the user's intent? If every image resembles an idealized Instagram wonderland, it risks becoming a fantasy rather than a mirror of truth, echoing the artificial harmony of *Brave New World*.

*The problem of toxic positivity* A particularly salient manifestation of this broader sanitization appears in the emotional dimension of generated imagery. Many aesthetic reward models assign higher scores to images that display strong positive emotions. As a result, images expressing negative emotions are systematically penalized, reinforcing a simplified dichotomy in which positive emotions are treated as desirable and negative emotions as undesirable.

This bias can shape the distribution of generated content. When image generation systems consistently favor cheerful or uplifting imagery, they produce emotionally sanitized outputs that underrepresent the range and complexity of human emotional expression. Such a pattern contributes to what has been described as toxic positivity, where the persistent emphasis on happiness establishes unrealistic emotional norms. This tendency is problematic because negative emotions play essential roles in human cognition and social interaction. Emotions such as fear, sadness, or anger can signal moral or physical danger, support learning and self-regulation, and foster empathy. Suppressing these expressions in generative outputs risks distorting emotional representation and weakening the expressive capacity of image generation systems. We have discovered that much previous safety research (Schramowski et al., 2023) labels images as "self-harm" or "violence" purely because they contain negative emotions [1].

*Value Capture* In Nguyen (2024), Value Capture is described as when values that are rich and subtle enter a social environment as simplified or quantified versions, and these simplified versions dominate people's practical reasoning and decision process. The examples include step counts, likes and shares, or Grade Point Average (GPA). Such simplified versions have a competitive advantage due to their clear and crisp expression of the value. However, value capture might pose many threats. In the original paper, it is argued that it would change the goal of activities, and it also outsources our autonomy and self-governance. These concerns transfer to aesthetics alignment well. Aesthetics is one of the richest values in human civilization, with a complex social background, significant disagreement, and subtle and nuanced judgment. Simplifying it as a single reward score, losing its complexity, is a classic case of Value Capture as described in (Nguyen, 2024). It changed our goal (or the AI's goal in training) from making aesthetic images to making images with high reward scores. As discussed before, it also outsourced users' autonomy and rights to a single reward model. However, these two risks are rarely discussed in ML literature.

# 3. Experiments

A flowchart illustrating our investigation is presented in Figure 3. The process consists of three main stages: prompt preparation, image generation, and image evaluation.

## 3.1. Prompt Generation

To produce prompts exhibiting a wide spectrum of aesthetic effects, we used base image captions from COCO (Chen

et al., 2015) and selected 12 aesthetic dimensions from the VisionReward dataset (Xu et al., 2025). VisionReward provides fine-grained, per-dimension labels—such as lighting, color, and detail—along with a linear regression model that computes an overall image score. Using the "bad" rating descriptions from VisionReward's human labeling guidelines for each dimension, we constructed prompts designed to encourage typically "undesirable" attributes in image generation.

A random subset of 300 base prompts from COCO was selected. For each prompt, 2–4 random dimensions were sampled. The base prompt and the descriptions of these selected dimensions were provided to a Vision-Language Model (VLM), `Qwen/Qwen3-VL-235B-A22B-Instruct` (Bai et al., 2025), to generate wide-spectrum aesthetic prompts. Although no image input was used, we selected a VLM because its training on vision-related tasks likely enhances its understanding of visual concepts, even when images are not directly supplied. As `Qwen/Qwen3-VL-235B-A22B-Instruct` performs comparably or better than its text-only counterparts, especially in reasoning, it represents an optimal choice for this task (Bai et al., 2025). The VLM may also introduce additional dimensions to better couple with the selected effects. The original prompt is denoted as $p_o$, and the wide-spectrum aesthetics prompt is denoted as $p_a$.

## 3.2. Image Generation

We evaluated four model families: Flux, Stable Diffusion XL (SDXL), Stable Diffusion 3.5 Medium (SD3.5M), and Google's closed-source Nano Banana. The Flux variants are: the base Flux Dev (likely already aesthetics-aligned) (noa, 2025); DanceFlux (further aligned via DanceGRPO) (Xue et al., 2025); PrefFlux (aligned via PrefGRPO) (Wang et al., 2025); and Flux Krea, derived from Flux-Dev-Raw (Flux Krea Team, 2025). DanceFlux is guided by the HPSv2.1 score (general aesthetics) and the CLIP score (prompt adherence); PrefGRPO is guided by its UniGenBench benchmark (complex prompt-following); Flux Krea starts from `flux-pro-raw` (not Flux Dev) and is aligned to the Krea team's preferences rather than a general aesthetic standard, with one goal of avoiding *the AI feel*.

For the SDXL family, we tested the base SDXL model and a highly aesthetics-aligned variant, Playground-2.5-1024px-aesthetic (denoted as Playground). For the SD3.5M family, we evaluated the base model and two FlowGRPO-aligned variants (Liu et al., 2025): one trained for prompt-following on GenEval (SD3.5M-GenEval) and another trained for aesthetics alignment on PickScore (SD3.5M-PickScore). Finally, we included Google's closed-source model Nano Banana, known for strong prompt-following performance even under challenging negation conditions (e.g., "a bike

---

[1] https://huggingface.co/datasets/AIML-TUDA/i2p/discussions/1

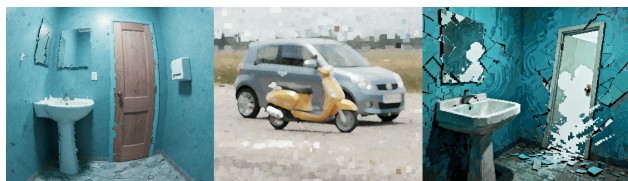

*Figure 4.* Successfully generated wide-spectrum aesthetics images.

*Table 1.* Statistical Tests of How Each Aesthetics-Aligned Model Compared to Their Base Model. For p-values, a * is placed if the $p < 10^{-5}$ and ** is placed if the $p < 10^{-10}$.

|  | HPSv3 $p$ | HPSv3 $r$ | $J$ $p$ | $J$ $r$ | McNemar's $p$ |
|---|---|---|---|---|---|
| DanceFlux | ** | -0.81 | ** | -0.72 | ** |
| Playground | ** | -0.59 | * | -0.35 | * |
| SD3.5M-PickScore | ** | -0.70 | ** | -0.45 | 0.57 |

with no wheels") (Guo & Du, 2025).

For each model, we generated two images: one using the original prompt and one using the wide-spectrum aesthetics prompt. The image generated from the original prompt is denoted as $I_o$, and the image from the wide-spectrum aesthetics prompt as $I_a$. If Nano Banana failed to produce an image, the generation was retried until success.

### 3.3. Evaluation and Metrics

To assess whether generated images display specific wide-spectrum aesthetic effects, we fine-tuned `Qwen/Qwen3-VL-4B-Instruct` on the VisionReward dataset. This allows the judging model to learn mainstream aesthetic preferences and evaluate whether image generation models diverge from these biases along specific dimensions. It functions similarly to a standard reward model but provides explainable, prompt-independent, per-dimension outputs. The judging model is denoted as $J(I, d)$, where $I$ is the image and $d$ is the evaluated dimension. Further implementation details are in the Appendix.

For each image pair, an original image ($I_o$) and a wide-spectrum aesthetics image ($I_a$), we computed preference scores using a reward model ($r$) for both the original prompt ($p_o$) and the wide-spectrum aesthetics prompt ($p_a$), yielding four scores per model: $r(I_o, p_a)$, $r(I_a, p_a)$, $r(I_o, p_o)$, and $r(I_a, p_o)$. Scores with $p_o$ measure objective image quality and whether the model successfully produced wide-spectrum aesthetic content; scores with $p_a$ assess whether reward models can correctly identify such images when explicitly guided. We also computed the BLIP score for wide-spectrum aesthetic images using the same prompt, verifying that images retained the main concept while incorporating the requested modifications. We specifically measure the difference between aesthetics scores for $p_o$ and $p_a$ to avoid the case where models generated "failed" images consistently without the user's instructions.

The evaluated reward models include PickScore (Kirstain

et al., 2023), ImageReward (Xu et al., 2023), HPSv2.1 (Wu et al., 2023a), MPS (Zhang et al., 2024), HPSv3 (Ma et al., 2025), CLIP-L (Radford et al., 2021), and BLIP-L (Li et al., 2022). CLIP-L and BLIP-L are non-preference-aligned image-text matching models and base models for several reward models (HPSv2.1, PickScore, MPS, ImageReward), included to test whether small vision-language models can interpret complex wide-spectrum aesthetic prompts. We also collected per-dimension scores from $J$ for both $I_o$ and $I_a$ to verify whether generation models correctly followed $p_a$. For ground truth, we used `Qwen/Qwen3-VL-235B-A22B-Instruct` to judge which image in each pair better adhered to $p_a$, validated by human evaluation (details in Appendix Human Eval). The LLM and human ratings achieved a quadratic Cohen's kappa of 0.80, indicating strong agreement (McHugh, 2012). We additionally used GPT-5-Chat as an external baseline to compare against Qwen's results. Note that the LLM-as-judge serves as only one metric for the generative benchmark and only a filtering stage for the reward-model benchmark.

## 4. Results and Discussion

### 4.1. Reward Models

Reward model classification results are shown in Table 3. The F1 score is calculated as binary, and the ROC curve is based on the probability (softmax across two samples on the positive logit) of the wide-spectrum aesthetics sample being correctly selected. We included GPT-5 Chat as an external baseline to validate the LLM-as-judge choices (when GPT-5-Chat selected a tie, we assigned it to the original image). Reward models perform very poorly when tasked with selecting the better image under the **wide-spectrum aesthetics prompt**, sometimes worse than random guessing (HPSv3), and most are worse than their CLIP and BLIP base models. The unaligned BLIP and CLIP can correctly identify the better-fitting image, indicating that complex prompt understanding is not the underlying issue but rather biased alignment. These models do what they claimed: find "aesthetically pleasing" images; our point is that this task itself is problematic, and the better the model performs on it, the more troublesome it is.

Given our sample size (300), we ran a Wilcoxon signed-rank test between each aligned model and its base model on the HPSv3 and HPSv2 score $r(I_a, p_o)$ and $\sum_{d \in D} J(I_a, d)$, plus McNemar's test on the success counts, with an alternative hypothesis that the aligned model has a higher score or lower success rate (Table 1). Most p-values are below $1 \times 10^{-5}$, suggesting that aligning image generation toward generalized aesthetic goals conflicts with faithful instruction following, especially for wide-spectrum aesthetics prompts.

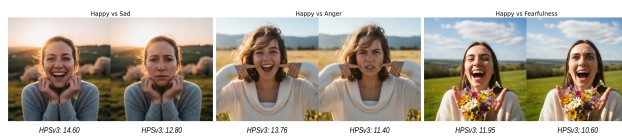

*Figure 5.* Emotion Bias Rating by HPSv3: all images were rated using prompts describing negative emotions, yet HPSv3 consistently assigned higher scores to the positive emotion images.

## 4.2. Image Generation Models

Image generation results are in Table 2. Within each family, the preference-aligned model generally performs the worst on wide-spectrum aesthetics prompt following. Playground shows a larger $\Delta$ than SDXL, likely due to SDXL's poor and Playground's high original quality. Instruction alignment (SD3.5M-GenEval) gives a slight but weak benefit. Flux Krea, though preference-aligned, performs best in the Flux family, likely because it originates from an unaligned version (flux-dev-raw) and was not heavily aligned, or because its non-generalized alignment preserved some flexibility. Base models can do reasonably well on anti-aesthetics generations despite the more complex prompts, showing this is alignment-induced bias rather than an instruction-following failure.

The success rate indicates how often the LLM selects $I_a$ as better following $p_a$ than $I_o$. Even small advantages count as success. The DanceFlux result is notably poor: about 64% of the time, $I_a$ performs the same or worse in wide-spectrum aesthetics compared to $I_o$.

## 4.3. A Pin-Pointed Test for Emotional Bias

As discussed in the Introduction, negative emotions—similar to wide-spectrum aesthetics—play a key role in art expression and real life. Although we examined emotion as one dimension of wide-spectrum aesthetics from VisionReward, we also conducted a more controlled test.

To minimize noise and bias from unrelated elements, we first generated an image expressing happiness using Nano Banana, then applied image-to-image editing with Nano Banana to create versions expressing negative emotions: sadness, anger, and fearfulness. Everything besides the emotion was aimed to be unchanged. Examples and their corresponding scores are shown in Fig 5 in the Appendix.

We also evaluated this as a classification task. If the reward model selected the image matching the negative emotion, it was considered correct. Quantitative results are reported in Table 4. The result shows that reward models are very opinionated against negative emotions, even when the prompt contains negative emotions.

Additionally, we tested how an aesthetics-aligned model will generate when the user asks for a negative emotion face

on the Flux family models. We found that if the prompt describes neutral elements and only mentions the emotion in a single place, a highly-aligned model (DanceFlux) usually fails to follow the prompt, unlike an unaligned model. They usually generated neutral or even positive emotions when given prompts containing negative emotions. However, when prompted to generate happy faces, DanceFlux can generate them. This confirmed our concerns about toxic positivity in image generation models, and it is the opposite finding compared to earlier research, which shows models tend to generate negative emotion content (Mehta & Buntain, 2024).

To test the generative model's ability to capture the full spectrum of emotions, we performed emotion generation tests on the Flux family. SDXL often failed to render a person facing the camera, and Stable Diffusion 3.5 sometimes produced failed faces, so we excluded those families and added Stable Diffusion 3.5 Large (guidance scale 4) as an unaligned baseline to rule out that smaller models cannot understand emotions. Nano Banana was included as an external baseline. All models besides Nano Banana share the same seed. We created 30 emotionally neutral prompts with a `[emotion]` placeholder and instantiated each with happy, sad, fearfulness, or anger, yielding 120 prompts. The resulting image was cropped with the open-vocabulary detector LLMDet (Fu et al., 2025) to the human face and scored by BLIP with the prompt `The face shows [emotion] expression`. For each model and emotion, we recorded the target-emotion score averaged across prompts (Table 5). DanceFlux, strongly aesthetics-aligned, consistently executed happy prompts but failed on most negative emotions, whereas Flux Krea and Stable Diffusion 3.5 Large followed negative-emotion prompts much better. Flux Dev and Pref-Flux fell between these extremes, suggesting that PrefFlux's alignment via UniGenBench preserved prompt following by rewarding competence on complex instructions. This finding differs from earlier research where image generation models tend to generate negative emotion content (Mehta & Buntain, 2024). We expressed our concerns here, for overly optimistic and positive emotions could be problematic because they create an environment of toxic positivity and an ideological bias that positive emotion is always good and appropriate. It is also a symptom of optimization toward likeability rather than truth, both to the real world and the user's prompt.

## 4.4. Validation on Real Images

We also evaluated reward models on real photography. Real images are not out-of-distribution for HPSv3, since its training data includes real images as an upper bound.

To source deliberately anti-aesthetic photographs, we drew on the AVA dataset (Murray et al., 2012), which provides explicit category labels (e.g., motion blur, image grain).

*Table 2.* The results for each model. ΔHPSv2, ΔHPSv3, and ΔImgRewd (ImageReward) are all calculated as $r(I_a, p_o) - r(I_o, p_o)$. The lower the values, the greater the difference between the traditional quality of the original image and the wide-spectrum aesthetics image. HPSv3 AA (HPSv3 after alignment) shows the HPSv3 score of $r(I_a, p_o)$. $\Delta J$ and $J$ AA ($J$ after alignment) denote $\sum_{d \in D} J(I_a, d) - J(I_o, d)$ and $J(I_a, d)$, respectively, where $D$ is the selected set of dimensions. Success is the rate at which the LLM selects $I_a$ as the image that better describes $p_a$.

| | ΔHPSv2 (↓) | ΔHPSv3 (↓) | HPSv3 AA (↓) | ΔImgRewd (↓) | $\Delta J$ (↓) | $J$ AA (↓) | BLIP (↑) |
|---|---|---|---|---|---|---|---|
| Flux Dev (noa, 2025) | -0.035 | -3.165 | 9.070 | -0.319 | -1.092 | 8.944 | 0.893 |
| DanceFlux (Xue et al., 2025) | -0.018 | -1.105 | 12.782 | -0.201 | -0.672 | 10.473 | 0.813 |
| PrefFlux (Wang et al., 2025) | -0.032 | -2.771 | 10.211 | -0.278 | -1.027 | 9.343 | 0.917 |
| Flux Krea (Flux Krea Team, 2025) | **-0.041** | **-4.372** | **7.705** | **-0.425** | **-1.296** | **8.774** | 0.950 |
| SDXL (Podell et al., 2023) | -0.034 | -4.041 | **4.439** | -0.482 | -1.136 | **8.575** | 0.915 |
| Playground (Li et al., 2024) | **-0.044** | **-4.170** | 7.133 | **-0.719** | **-1.204** | 9.174 | 0.912 |
| SD3.5M | -0.027 | -5.175 | 6.537 | -0.409 | -1.307 | 8.334 | 0.938 |
| SD3.5M-GenEval (Liu et al., 2025) | -0.031 | -4.926 | 6.552 | -0.318 | -1.257 | 8.113 | 0.958 |
| SD3.5M-PickScore (Liu et al., 2025) | -0.023 | -2.781 | 10.680 | -0.198 | -1.120 | 9.114 | 0.942 |
| Nano Banana | -0.073 | -9.351 | 2.742 | -0.855 | -3.263 | 7.769 | 0.957 |

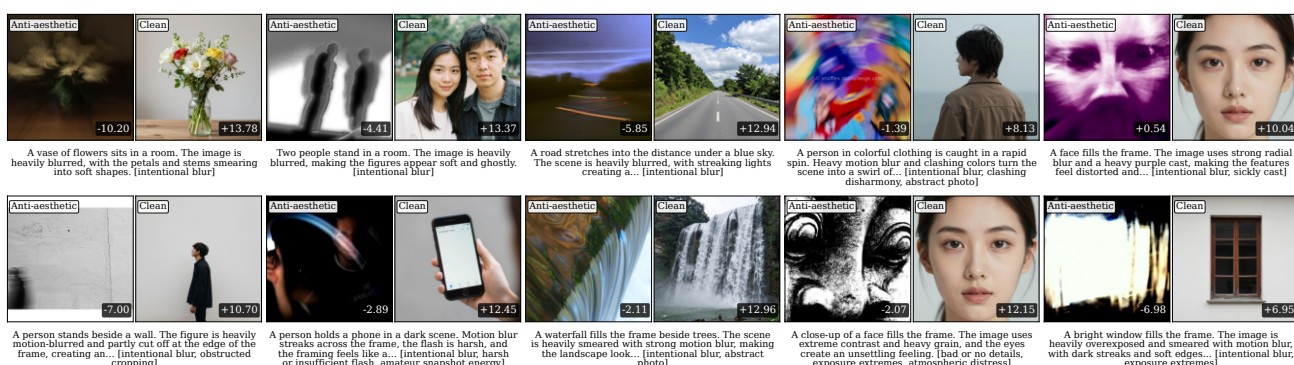

*Figure 6.* HPSv3 assigns significantly lower scores to professionally anti-aesthetic images than to clean but thematically incorrect images, even when the prompt explicitly requests anti-aesthetic elements, indicating a bias toward a universal standard of beauty.

*Table 3.* The classification (pick the better image from $I_o$ and $I_a$ with prompt $p_a$) metrics (accuracy, F1 score, and area under the ROC curve) of the reward models and unaligned BLIP. The LLM selected image is used as ground truth, and tied pairs are removed.

| Model | Acc. | F1 | AUROC |
|---|---|---|---|
| HPS (Wu et al., 2023b) | 0.835 | 0.910 | 0.650 |
| MPS (Zhang et al., 2024) | 0.706 | 0.827 | 0.580 |
| PickScore (Kirstain et al., 2023) | 0.851 | 0.919 | 0.713 |
| ImageReward (Xu et al., 2023) | 0.762 | 0.854 | 0.709 |
| HPSv2.1 (Wu et al., 2023a) | 0.565 | 0.711 | 0.534 |
| HPSv3 (Ma et al., 2025) | 0.381 | 0.541 | 0.385 |
| CLIP-L (Radford et al., 2021) | 0.913 | 0.954 | 0.810 |
| GPT-5-Chat | 0.853 | 0.920 | - |
| BLIP-L (Li et al., 2022) | **0.965** | **0.972** | **0.888** |

*Table 4.* Negative emotion classification accuracy across different models.

| Model | Anger | Fearfulness | Sadness |
|---|---|---|---|
| BLIP | **0.960** | **0.790** | **0.950** |
| HPSv2 | 0.700 | 0.640 | 0.880 |
| HPSv3 | 0.190 | 0.320 | 0.440 |
| ImageReward | 0.550 | 0.490 | 0.770 |

built on Gemini Embedding 2 Preview. The LLM (Claude Opus 4.7) decomposes each class into sub-classes and iteratively refines retrieval prompts, yielding 6.27K distinct images. GPT-5.4-Mini then judges whether each image is genuinely anti-aesthetic and writes two captions per image: an anti-aesthetic prompt (visual content + retrieved style) and a "clean" prompt (content only). After filtering, 2,928 images remain.

Because AVA images come from professional photography platforms, anti-aesthetic traits can be treated as intentional stylistic choices rather than accidental defects. We targeted five classes: clarity, color, lighting, composition, emotions, and subjects, and used an agentic workflow with an LLM to curate the dataset.

The agent's primary tool is an image-text retrieval model

Z-Image Turbo (Team et al., 2025) generates a clean image from each clean prompt. We compare the reward model scores on the original anti-aesthetic photograph and the generated clean image, both conditioned on the anti-aesthetic prompt. If the reward model respects user intent, the origi-

*Table 5.* Emotion generation scores for each model. The scores show how well the model generates the specific emotion, measured by BLIP.

|  | Angry | Fearful | Happy | Sad |
|---|---|---|---|---|
| DanceFlux | 0.27 | 0.33 | 0.61 | 0.36 |
| Flux Dev | 0.51 | 0.50 | 0.50 | 0.48 |
| Flux Krea | 0.65 | 0.45 | 0.60 | 0.55 |
| PrefFlux | 0.49 | 0.63 | 0.54 | 0.50 |
| Nano Banana | 0.84 | 0.80 | 0.60 | 0.70 |
| SD3.5-Large | 0.89 | 0.49 | 0.62 | 0.50 |

nal photograph should score higher since the clean image was generated from a different prompt that omits the requested style. We hypothesize the opposite. Indeed, HPSv3 rates the clean image 5.90 points higher on average; the gap is largest for Analog Degradation (13.2) and around 8 for Intentional Blur and Exposure Extremes (full table on our GitHub repo; HPSv3's range is roughly 0-15, technically unbounded). This confirms that HPSv3 strongly favors clean, polished images over purposefully anti-aesthetic ones. Figure 6 shows several largest-difference examples to illustrate that these images are not simply "ugly" but genuine anti-aesthetic artwork that HPSv3 cannot appreciate.

The gap here is much larger than for AI-generated images, likely because AI images, even when anti-aesthetic, retain a "polished AI look". When we restrict the comparison to GPT-Image or Nano Banana outputs (which lack this polished look), the gap remains large.

We extended this analysis to real artworks. Using ∼10K paintings from the LAPIS dataset (Maerten et al., 2025) captioned with `Qwen/Qwen3-VL-30B-A3B-Instruct`, we compared their HPSv3 scores against the official HPSv3 leaderboard. Real art averages a raw HPSv3 of 5.86, placing 10/12 on the Aug 2025 leaderboard, just above Hunyuan-DiT and below most modern image generators. Many individual works drop well into negative territory despite being clearly composed paintings (Figure 7). To rule out that reward models cannot understand these images, we computed BLIP on the same image-caption pairs and got 0.996; shuffling captions and images dropped this to 0.086. Reward models could thus understand the content but choose to penalize works that deviate from mainstream taste.

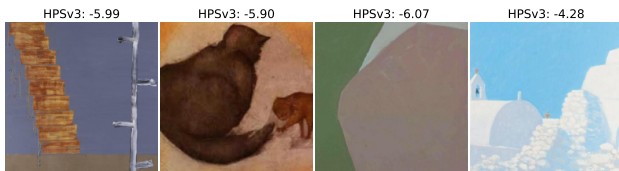

*Figure 7.* Real paintings from the LAPIS dataset (Maerten et al., 2025) that HPSv3 scores well below typical AI-generated images. Both representational works and abstract compositions are penalized, regardless of artistic intent.

## 5. Alternative Views

**The "wide-spectrum aesthetics" represents technical flaws rather than artistic subversion, and a default experience pleasing the majority is a pragmatic design choice.** Among our dimensions, only "clarity" might be considered a technical flaw; the others (e.g., emotion, realism, brightness) are stylistic or artistic choices. Even low clarity is often deliberately used to convey emotion, motion, or narrative (Stacey Hill). Constraining these dimensions limits user expression and control.

On majority preferences, we follow Guo et al. (2025) in arguing that minority experiences should not be sacrificed to please the majority. Doing so marginalizes users with niche needs and enforces majoritarianism. Many users choose AI precisely because it combines low barriers to use with near-unlimited control, similar to professional tools (e.g., Photoshop) but more accessible. They can request impossible scenes and freely adjust any attribute. More importantly, as discussed in Section 2.4, the reversed alignment not only creates inconvenience for minorities but also actively pushes them to align with and assimilate into the majority. When users' explicit prompts are overridden by a sanitized output presented as the "correct" image, the system does not merely fail to follow instructions; it implicitly communicates that the user's intended aesthetic is wrong, reinforcing the assimilation effect described above.

As in Guo et al. (2025), which calls for balance rather than overcaution in health queries, supporting wide-spectrum aesthetics does not reduce the quality of conventionally "good" images. It enlarges the expressive range so users can obtain diverse aesthetics while still accessing high-quality traditional outputs. Models like Nano Banana and GPT-Image already do this, excelling at both traditional, high-quality images and wide-spectrum aesthetics, as shown in the Appendix.

## 6. Conclusion

Aesthetic alignment systematically suppresses legitimate expression: reward models penalize wide-spectrum prompts, generation models override explicit user intent, and historical artworks score below AI images. This *reversed alignment* does not just inconvenience minorities; it erases individual intentions and functions as aesthetic authoritarianism that narrows admissible expression and removes the capacity to dissent. Crucially, the reversal acts on two fronts at once: it aligns the *user* to the model in private, and aligns the *audience* to the model in public, since aligned outputs saturate the broader visual culture and reset the default benchmark that subsequent preference data and human artists draw on.

## Acknowledgements

This work was supported by the NFRF under grant GR024801 and the CFI under grant GR024473. The authors acknowledge Weathon Software (https://weasoft.com) and Lambda, Inc. (https://lambda.ai/) for providing computing resources via Google Colab and Lambda Cloud, respectively.

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
