# OpenReview forum: "Position: Universal Aesthetic Alignment Narrows Artistic Expression"
_ICML.cc/2026/Position_Paper_Track — ICML 2026 Position Paper Track spotlight_

### Official Review · Reviewer_xPeF · 2026-02-16

**Significance:** 3
**Argument Clarity:** 3
**Rating:** 4
**Confidence:** 3

**Questions:**

- Have you measured the length/complexity difference between wide-spectrum and original prompts?
- How sensitive are your results to the choice of LLM judge or its prompt?
- Do you have evidence of actual user demand for anti-aesthetic outputs (e.g., user studies)?

**Alternative Views Section:**

Yes

**Compliance With Llm Reviewing Policy A Conservative:**

Affirmed.

**Discussion Potential:**

3

**Final Justification:**

My concerns are addressed by the rebuttal. Raising score.

**Paper Summary:**

This paper argues that aesthetic alignment in image generation suppresses artistic expression. Moreover, it overrides user intent. Through experiments on multiple generations and reward models using anti-aesthetic prompts, the authors show that aligned models default to conventionally beautiful outputs despite explicit instructions otherwise, and reward models penalize prompt-faithful anti-aesthetic images.

**Position:**

Yes

**Position In Title:**

Yes

**Related Work:**

3

**Strengths And Weaknesses:**

Strengths:
- The tension between aesthetic alignment and instruction fidelity is an interesting and important topic.
- The five-risk taxonomy in Section 2.4 provides a clear conceptual framework grounded in prior work.
- The paper evaluates numerous generation and reward models across text-to-image, image-to-image, real artwork, and emotion-specific experiments.

Weaknesses:
- There are only 300 COCO-derived prompts. This may limit generalizability.
- Anti-aesthetic prompts are longer and more complex than originals. Therefore, it could be hard to disentangle aesthetic over-alignment from prompt difficulty.
- Ground truth relies primarily on VLMs. However, these VLMs themselves may also have aesthetic biases. In the meantime, human validation covers only 40 images.

**Support:**

3

---

> ### Author Rebuttal · Authors · 2026-03-27
>
> We want to thank the reviewer for the comments.
>
> > Have you measured the length/complexity difference between wide-spectrum and original prompts?
>
> We have addressed this concern in Table 2, where we compared the base model, the aesthetically aligned model, and the instruction-aligned model on the same anti-aesthetic prompts. Both the base model and the instruction-aligned model outperform the aesthetically aligned model. Since all three models receive identical prompts, any performance gap cannot be attributed to prompt complexity. This isolates aesthetic alignment as the cause: post-training in general does not compromise anti-aesthetic ability, but aesthetic alignment specifically does.
>
> > How sensitive are your results to the choice of LLM judge or its prompt?
>
> We cross-validated our results from Qwen-3 using GPT-5, achieving acc=0.853 and f1=0.920. Since the LLM is asked which image better fits the prompt rather than which looks better, its own aesthetic bias is unlikely to contribute. We further analyzed divergence cases between the LLM judge and human annotators: disagreements occur exclusively on borderline near-tie cases rather than systematically across the distribution. One example is shown here: https://i.imgur.com/TjDERC0.jpeg. We also emphasize that VLM judgment is only one of our multiple metrics.
>
> > Do you have evidence of actual user demand for anti-aesthetic outputs (e.g., user studies)?
>
> Deliberate deviation from mainstream aesthetics is a historically continuous creative tradition with real practitioners. Professional photography communities such as dpchallenge.com actively celebrate intentionally blurred, overexposed, and underexposed images (examples: https://imgur.com/a/KAmaDOt), and the CVF art exhibition (thecvf-art.com) curates works exploring glitch aesthetics and non-photorealistic styles. The question of majority demand is therefore beside the point: these are established creative modes, and aesthetic alignment should not structurally suppress them.
>
>
> > There are only 300 COCO-derived prompts. This may limit generalizability.
>
> We select three representative models spanning different alignment strengths (Flux Krea, DanceFlux, Flux Dev) and validate the trend on 1000 anti-aesthetics prompts. This matches the trend in our main results, where DanceFlux (most strongly aligned) has the poorest anti-aesthetics capability, while Flux Krea successfully produces anti-aesthetic elements. A lower HPSv3 score means better anti-aesthetics generation.
>
> | model | HPSv3 Score |
> |:---|---:|
> | dance | 12.0908 |
> | dev | 9.99115 |
> | krea | 8.9572 |
>
> We also noticed that the written review does not mention any specific concerns about our related work coverage, but the score for this criterion is 1 (poor). We would greatly appreciate clarification on what is missing or insufficient, so we can address it properly in the final version.

---

> > ### Author Rebuttal · Reviewer_xPeF · 2026-04-03
> >
> > Rebuttal addresses my concerns. The related work score of 1 was my oversight. Willing to raise my score.

---

### Official Review · Reviewer_5ZVE · 2026-03-13

**Significance:** 3
**Argument Clarity:** 3
**Rating:** 5
**Confidence:** 5

**Questions:**

Please refer to the Weaknesses. If the authors can address these concerns, I would be willing to increase my score.

**Alternative Views Section:**

Yes

**Compliance With Llm Reviewing Policy A Conservative:**

Affirmed.

**Discussion Potential:**

3

**Final Justification:**

The rebuttal has satisfactorily addressed my concerns, and I will therefore increase my score to Accept.

**Paper Summary:**

This paper argues that current image generation systems align with a single dominant aesthetic preference, which may conflict with the user's actual intent and restrict legitimate creative expression. The authors introduce the concept of broad-spectrum aesthetics to describe outputs that intentionally deviate from mainstream aesthetic norms. They show that existing reward models and aligned generation models often penalize such outputs, even when the user explicitly requests them. The paper further argues that aesthetic alignment encodes developer-driven aesthetic standards within the system, which weakens aesthetic diversity. Based on these observations, the authors advocate that future alignment strategies should place greater emphasis on instruction faithfulness and support diverse aesthetic expression.

**Position:**

Yes

**Position In Title:**

Yes

**Related Work:**

2

**Strengths And Weaknesses:**

**Strengths:**

1. The paper raises a clear and concrete position that aesthetic alignment in image generation systems may affect user freedom of expression and creative control.

2. The concept of broad spectrum aesthetics provides a unified framework for discussing outputs that deviate from mainstream aesthetic norms.

3. The argument is supported by analyses of multiple models and evaluation tools, which strengthens the overall reasoning.

**Weaknesses:**

1. The paper interprets aesthetic alignment mainly as developer dominance. It provides a limited analysis of technical factors that may produce similar effects, such as training data distribution or annotation procedures.

2. The paper does not discuss whether the main idea from fairness-oriented generation [1,2,3,4] should be considered. In such works, when aesthetic standards are not specified, images of different styles are generated with equal probability. The authors should discuss this issue.

3. The paper assumes that user intent should take priority over general preferences. It does not sufficiently examine cases where shared aesthetic norms may provide reasonable constraints.

[1] Fair Diffusion: Instructing text-to-image generation models on fairness.

[2] Unified Concept Editing in Diffusion Models.

[3] Finetuning Text-to-image Diffusion Models for Fairness.

[4] LightFair: Towards an Efficient Alternative for Fair T2I Diffusion via Debiasing Pre-trained Text Encoders.

**Support:**

2

---

> ### Author Rebuttal · Authors · 2026-03-27
>
> We want to thank the reviewer for the comments.
> > The paper interprets aesthetic alignment mainly as developer dominance. It provides a limited analysis of technical factors that may produce similar effects, such as training data distribution or annotation procedures.
>
> We would like to clarify that the issue is not pre-training data bias but RLHF data bias. Our experiments demonstrate this directly: base models without aesthetic alignment can produce anti-aesthetic images faithfully, showing that the pretraining data itself does not suppress this capability. The problem is introduced specifically during the alignment stage, which is defined and controlled by developers. Training data (in RLHF) bias and developer intent are therefore two sides of the same coin, annotators are recruited by developers, data is filtered by developers, and annotation policies are set by developers. Developer intent is the root cause, and data bias is how it manifests in RLHF pipelines.
> HPSv3’s self-report illustrates this concretely. The annotator pool is already skewed toward a narrow demographic, with 88.95% aged between 18 and 40, and the 21-30 age group alone accounting for 38.65%. The pipeline then compounds this through structural filtering: annotators must pass a proficiency gate requiring 80% convergence with professional artists, and only image pairs exceeding 95% inter-annotator confidence are used for training. This design explicitly discards disagreement, which is precisely where unconventional and anti-aesthetic preferences would appear. Furthermore, HPSv3 designates real photographs as the upper bound of image quality, encoding a realism bias that treats photorealism as the gold standard. This structurally penalizes stylized, abstract, or anti-aesthetic work regardless of its artistic intent or technical execution. Taken together, these are not incidental data artifacts — they are developer design decisions that systematically narrow what counts as "good" output before training begins.
>
> > The paper does not discuss whether the main idea from fairness-oriented generation [1,2,3,4] should be considered. In such works, when aesthetic standards are not specified, images of different styles are generated with equal probability. The authors should discuss this issue.
>
> Since ICML does not allow PDF updates during rebuttal, we will incorporate this literature in the final version around line 085. Our paper already discusses demographic bias in generation (Algorithmic Gaze, AesBiasBench), and we will add the additional references the reviewer suggested. To the best of our knowledge, no prior work connects this bias to aesthetic alignment itself, which is the core contribution of our paper.
> Fairness-oriented generation [1,2,3,4] addresses a related but structurally distinct problem. These works aim to equalize demographic representation in model outputs, operating with relatively clear normative targets such as parity across protected attributes. Aesthetic preference, by contrast, has no such ground truth — there is no principled way to define a "fair" distribution over aesthetic styles, because what counts as a valid style is itself contested. We therefore frame our problem around instruction faithfulness rather than distributional fairness: the goal is not to equalize representation across styles, but to ensure models do not override explicit user requests based on internalized aesthetic norms.
>
>
> > The paper assumes that user intent should take priority over general preferences. It does not sufficiently examine cases where shared aesthetic norms may provide reasonable constraints.
>
> We agree that shared aesthetic norms serve legitimate functions. For casual users with underspecified prompts, conventional defaults can provide a good user experience. In functional domains such as medical illustration, adherence to visual standards ensures clarity. In organizational settings, pre-established visual standards (e.g., brand guidelines) may also reasonably constrain outputs for consistency.
> However, these are better understood as contextual defaults, not as justification for model-level aesthetic override. An organization enforcing its own style guide exercises aesthetic authority over its own outputs; a model silently penalizing user-requested styles exercises the developer's authority over the user's outputs. Our position is that defaults should remain overridable when the user explicitly communicates otherwise. We are not aware of a compelling case where a model should refuse a clear aesthetic instruction on purely aesthetic, non-safety grounds. Content safety constraints are a separate and necessary baseline but should not extend to suppressing provocative commentary, negative emotional expression, or unconventional artistic intent.

---

> > ### Author Rebuttal · Reviewer_5ZVE · 2026-04-04
> >
> > I have carefully reviewed the authors' responses to my comments, together with the feedback from the other reviewers. The rebuttal has satisfactorily addressed my concerns, and I will therefore increase my score to Accept.

---

### Official Review · Reviewer_83Qz · 2026-03-13

**Significance:** 3
**Argument Clarity:** 3
**Rating:** 4
**Confidence:** 3

**Questions:**

No further questions. Please see the weakenesses.

**Alternative Views Section:**

Yes

**Compliance With Llm Reviewing Policy A Conservative:**

Affirmed.

**Discussion Potential:**

3

**Final Justification:**

Overall, the designed procedure is effective to support the proposed position. Most of my concerns have been addressed during the rebuttal. I would like to keep my initial score, which is already an accept.

**Paper Summary:**

This work focus on a critical problem: the image generation models are over-aligned to a generalized aesthetic preference, which conficts with user intent when anti-aesthetic outputs are required. It constructs a dataset and a pipeline to evaluate state-of-the-art generation and reward models for the purpose of demonstrating the proposed position. Briefly, it uses base image captions from COCO and a large VLM to produce prompts exhibiting a wide spectrum of aesthetic effects. Then, it sends the prompts to mainstream image generation to create images. Next, it introduces some metrics wo asess whether the generated images display specific wide-spectrum aesthetic effects.

**Position:**

Yes

**Position In Title:**

Yes

**Related Work:**

3

**Strengths And Weaknesses:**

Strengths:
- The focused problem is very important and will have great impact for the future development of image generation models.
- The designed procedure is effective to provide experimental evidence for the proposed position.
- The experimental results are comprehensive and well-organized.

Weaknesses:
- Deeper insights and analysis are expected. Currently this work reveals a phenomenon. This phenomenon seems to be a common sense in the research community, while I acknowledge the contribution that this work tries to quantify such phenomenon. It would be better if there are some discussion about root cause and potential solution, or suggestions for common users.
- The sample size is relatively small and the coverage of the investigated image generation model could be further extended, e.g., GPT-image-gen, Qwen-Image, etc.

**Support:**

3

---

> ### Author Rebuttal · Authors · 2026-03-27
>
> We thank the reviewer for the positive assessment and address the weaknesses below.
>
> 1. We agree that moving beyond phenomenon quantification is valuable. On the root cause side, we argue the issue lies specifically in the Reinforcement Learning from Human Feedback (RLHF) alignment stage, rather than pretraining. Our experiments show that base models can produce anti-aesthetic outputs relatively faithfully, meaning the capability exists but is suppressed during alignment. The alignment pipeline itself encodes developer-defined aesthetic preferences through annotator selection.
>
> HPSv3’s self-report illustrates this concretely. The annotator pool is already skewed toward a narrow demographic, with 88.95% aged between 18 and 40, and the 21-30 age group alone accounting for 38.65%. The pipeline then compounds this through structural filtering: annotators must pass a proficiency gate requiring 80% convergence with professional artists, and only image pairs exceeding 95% inter-annotator confidence are used for training. This design explicitly discards disagreement, which is precisely where unconventional and anti-aesthetic preferences would appear. Furthermore, HPSv3 designates real photographs as the upper bound of image quality, encoding a realism bias that treats photorealism as the gold standard. This structurally penalizes stylized, abstract, or anti-aesthetic work regardless of its artistic intent or technical execution.
>
>
> 2. On the solutions side, we discuss several directions in the paper. We call for a solution that reduces the alignment strength in the base model but shifts the aesthetics/anti-aesthetics preference to users by prompting or specific parameters. Our results also show that some models, notably GPT-Image and Nano Banana, already handle aesthetic and anti-aesthetic outputs simultaneously (Table 2, Figure 15), suggesting the problem is tractable. However, these methods are closed source. For open source alternatives, we discuss the negative guidance method in Appendix Section I and Table 13. By using NAG or VSF, we can successfully move away from the "generalized beauty standard". Figure 14 demonstrates a tunable LoRA distilled from NAG that acts as a lightweight, practical solution. We could train another LoRA on professional anti-aesthetics images from professional websites (that are actually "aesthetically ugly" and not by technical mistakes) to mix with the AI-generated data for more diverse images. The query is done by an agentic process. An anonymized preview of our real-image dataset is available at https://imgur.com/a/KAmaDOt, showing samples spanning multiple anti-aesthetic categories (intentional blur, exposure extremes, flat contrast, film artifacts) with a wide HPSv3 score range. We also confirmed that these images are ranked low on the HPSv3 benchmark even when given anti-aesthetics prompts, confirming that the images collected are anti-aesthetics. Leaderboard: https://imgur.com/a/DDxUGG6
>
> 3. On sample size and model coverage. We have extended our evaluation to include Qwen-Image, GPT-Image, and SeedDream 4 on the same 300-prompt dataset. We want to thank the reviewer for proposing testing GPT-Image, as the results are notably strong; it produced "aesthetically ugly" or "anti-aesthetics" images very well (anonymized preview: https://imgur.com/a/5kqANc0). Results are shown in the table below. We can see that GPT-Image has great anti-aesthetics scores, yet at the same time keeps a high original aesthetics score when the user requests normal images.
>
> | model | HPSv3 Original | HPSv3 Anti Aesthetics | Delta |
> |:---|---:|---:|---:|
> | gpt-image-1.5 | 13.3244 | -1.17499 | -14.5 |
> | qwen_image | 12.4953 | 7.66325 | -4.8 |
> | seeddream4 | 11.772 | 5.2101 | -6.6 |
>
> 4. For the dataset, we select three representative models spanning different alignment strengths (Flux Krea, DanceFlux, Flux Dev) and validate the trend on 1000 more anti-aesthetics prompts. This matches the trend in our main results, where DanceFlux (being aligned strongest) has the poorest anti-aesthetics capability, while Flux Krea, which avoided aligning to a general preference, successfully provided anti-aesthetics elements. A lower HPSv3 score means better anti-aesthetics generation.
>
> | model | HPSv3 Score |
> |:---|---:|
> | dance | 12.0908 |
> | dev | 9.99115 |
> | krea | 8.9572 |

---

> > ### Author Rebuttal · Reviewer_83Qz · 2026-04-03
> >
> > Thank the authors for their feedbacks. I have no further questions.

---

### Official Review · Reviewer_e8y7 · 2026-03-15

**Significance:** 3
**Argument Clarity:** 3
**Rating:** 5
**Confidence:** 4

**Questions:**

See weaknesses.

**Alternative Views Section:**

Yes

**Compliance With Llm Reviewing Policy A Conservative:**

Affirmed.

**Discussion Potential:**

4

**Final Justification:**

The experiments in this paper are real and the results clearly support the position. The rebuttal clarified that the authors aren't anti-alignment but pro-controllability, which I think is the right framing. Both concerns addressed. Keeping my score at 5.

**Paper Summary:**

This paper argues over-aligning image generation models to a generalized aesthetic preference actively harms artistic expression, especially for "wide-spectrum aesthetics". Authors build a wide-spectrum aesthetics dataset, test multiple generation models and reward models. Key findings include that preference-aligned models perform worst on wide-spectrum prompts.

**Position:**

Yes

**Position In Title:**

Yes

**Related Work:**

4

**Strengths And Weaknesses:**

Strengths
1. I think the biggest thing going for this paper is that it actually runs experiments. Multiple model families, multiple reward models, human validation, and several distinct experimental setups. That's a lot of work for a position paper and honestly it shows.
2. The result that unaligned CLIP-L and BLIP-L outperform all preference-aligned reward models at identifying wide-spectrum aesthetic images really stuck with me. It's not that the models can't understand these images, it's that alignment actively breaks their understanding. I thought that was a nice finding.
3. Figure 4 where The Scream gets 5.23 on HPSv3 while generic AI images score 10+ is just kind of absurd, and I think it makes the point better than any amount of argumentation could. The 10K LAPIS evaluation adds scale to this.

Weaknesses
1. I think the Alternative Views section is a bit one-sided. I would've liked to see the authors engage more seriously with the idea that alignment is a practical safety/quality measure for most users, and that maybe the fix is user-controllable alignment strength rather than scrapping alignment altogether.
2. The LLM judging pipeline (Qwen3-VL) does a lot of heavy lifting. Kappa of 0.80 is fine but I'm curious about what kinds of cases the LLM and humans disagreed on.

**Support:**

4

---

> ### Author Rebuttal · Authors · 2026-03-27
>
> We want to thank the reviewer for the comments.
>
> On the Alternative Views section.
> We appreciate this suggestion and agree that user-controllable alignment strength is a practical and desirable direction. This is in fact, consistent with our proposed solutions. Figure 14 (in Appendix) demonstrates a tunable LoRA approach, which can be naturally extended to support both an aesthetics LoRA and an anti-aesthetics LoRA on top of an untouched base model, giving users direct control over aesthetics alignment strength and direction. We will also include in the future work section that something similar to AttriCtrl [1] would be beneficial, where the user can set specific dimensions of attributes. Alternatively, the model could be aligned with high text fidelity (instead of “general aesthetics”) where it can default to universal beautiful images but also divate it when the user requests directly using a prompt.
> Our results show that GPT-Image and Nano Banana already achieve high fidelity to user intent regardless of whether the request is aesthetic or anti-aesthetic (Figure 15 in the appendix). We are not arguing for the removal of aesthetic alignment but for making it a controllable default rather than an immutable constraint, which is precisely what these solutions demonstrate.
>
>
> On LLM, judge disagreement cases.
>
> We agree this is worth examining. Since the measure is a simple “which image better fits the prompt” question, we suspect that these cases are on the edge where either left or right is a good fit. We have sampled these samples and confirmed this. Since ICML does not allow inserted images in rebuttal, we put one example in this anonymized link: https://i.imgur.com/TjDERC0.jpeg. We also want to emphasize that VLM judgment is only one of our many metrics used.
>
> [1] https://arxiv.org/abs/2508.02151

---

> > ### Author Rebuttal · Reviewer_e8y7 · 2026-04-03
> >
> > Thanks for the response. The tunable LoRA idea and the pointer to GPT-Image/Nano Banana already achieving high text fidelity are good answers to W1. I agree that the goal should be controllable alignment, not removing it. W2 edge-case explanation makes sense given the task is a simple preference question.

---

### Decision · Program_Chairs · 2026-04-30

**Decision:**

Accept (spotlight)

**Comment:**

This paper was reviewed by four expert reviewers, receiving uniformly positive ratings (2x Accept, 2x Borderline Accept). All reviewers acknowledged the importance of the problem and appreciated the experimental rigor. Initial concerns centered on the Alternative Views section, sample size, root cause analysis, and prompt complexity effects. The rebuttal phase was effective. Authors provided additional experiments including GPT-Image and Qwen-Image evaluations, extended the dataset to 1000 prompts for validation, clarified that the issue stems from RLHF rather than pretraining data, and demonstrated concrete solutions via tunable LoRAs. All reviewers explicitly confirmed their concerns were resolved, with Rev#5ZVE and xPeF raising their scores to Accept.

Rev#e8y7 particularly valued the experimental depth and the finding that CLIP/BLIP outperform preference-aligned models on wide-spectrum aesthetics. Rev#83Qz noted the effective experimental design supporting the position. Rev#5ZVE appreciated the engagement with fairness literature and the instruction faithfulness framing. Rev#xPeF's concerns about prompt complexity and generalizability were addressed through controlled comparisons and extended validation.

The paper makes a timely contribution with clear practical implications for model development. The experimental evidence is good, covering multiple model families, reward models, image-to-image editing and real artwork evaluation. The proposed solutions (controllable alignment via LoRAs, instruction-fidelity prioritization) are concrete and technically grounded. The paper is recommended for acceptance.